# Comparison of Tacrolimus Intra-Patient Variability during 6–12 Months after Kidney Transplantation between CYP3A5 Expressers and Nonexpressers

**DOI:** 10.3390/jcm11216320

**Published:** 2022-10-26

**Authors:** Almas Nuchjumroon, Somratai Vadcharavivad, Wanchana Singhan, Manorom Poosoonthornsri, Wiwat Chancharoenthana, Suwasin Udomkarnjananun, Natavudh Townamchai, Yingyos Avihingsanon, Kearkiat Praditpornsilpa, Somchai Eiam-Ong

**Affiliations:** 1Department of Pharmacy Practice, Faculty of Pharmaceutical Sciences, Chulalongkorn University, Bangkok 10330, Thailand; 2Department of Pharmaceutical Care, Faculty of Pharmacy, Chiang Mai University, Chiang Mai 50200, Thailand; 3Pharmacy Department, King Chulalongkorn Memorial Hospital, The Thai Red Cross Society, Bangkok 10330, Thailand; 4Tropical Immunology and Translational Research Unit, Department of Clinical Tropical Medicine, Faculty of Tropical Medicine, Mahidol University, Bangkok 10400, Thailand; 5Department of Medicine, Faculty of Medicine, Chulalongkorn University, Bangkok 10330, Thailand

**Keywords:** CYP3A5, genetic polymorphism, immunosuppression, intra-patient variability, kidney transplantation, tacrolimus, therapeutic drug monitoring

## Abstract

A high intra-patient variability (IPV) of tacrolimus exposure is associated with poor long-term kidney transplantation outcomes. To assess the influence of *cytochrome P450 (CYP) 3A5* genetic polymorphisms on tacrolimus IPV, 188 clinically stable kidney transplant recipients, who had received an immediate-release tacrolimus-based immunosuppressive regimen, were enrolled in this retrospective cohort study. Genotyping of *CYP3A5*3* (rs776746) was performed and 110 (58.5%) were identified as CYP3A5 expressers and 78 (41.5%) as nonexpressers. Whole blood tacrolimus concentrations were analyzed by chemiluminescent microparticle immunoassay. Dose-adjusted trough tacrolimus concentrations (C0/D) measured at months 6, 9, and 12 were used to determine IPV. There were no significant differences in the IPV estimated by the coefficient of variation, the IPV calculated by mean absolute deviation method, and the proportions of recipients with the IPV estimated by the coefficient of variation of 30% or more between CYP3A5 expressers and nonexpressers (*p* = 0.613, 0.686, and 0.954, respectively). Tacrolimus C0/D in CYP3A5 expressers was approximately half of those in nonexpressers, overall (*p* < 0.001). In both CYP3A5 expressers and nonexpressers, tacrolimus C0/D increased gradually from month 6 to month 12 (*p* = 0.021). There was no evidence that the *CYP3A5* polymorphisms significantly influence tacrolimus IPV during the 6 to 12 months after kidney transplantation.

## 1. Introduction

Tacrolimus is a potent calcineurin inhibitor that is widely used as the cornerstone of medication in the maintenance of immunosuppressive therapy for the prevention of acute allograft rejection after kidney transplantation (KT) [1]. The benefits of a tacrolimus-based regimen in improving KT outcomes have been demonstrated [2,3,4].

Tacrolimus possesses a narrow therapeutic window. Underexposure to tacrolimus increases the risk of rejection and kidney graft loss, whereas overexposure is associated with an enhanced risk of infection and toxicity [5]. In addition, tacrolimus exhibits substantial pharmacokinetic variability. The dose required to reach the same target tacrolimus concentrations varies considerably between individual patients and within an individual patient over time [5]. In kidney transplant recipients (KTRs), tacrolimus therapy is routinely personalized by therapeutic drug monitoring. In clinical practice, whole blood trough concentration (C0) of tacrolimus is typically monitored with a subsequent dosage adjustment [6].

Intra-patient variability (IPV) of tacrolimus exposure, which is defined as the fluctuation in tacrolimus blood concentrations within an individual over a certain period during which the tacrolimus dose is left unchanged [7], has garnered attention, given a growing body of evidence that suggests high IPV is associated with poor long-term outcomes, including acute rejection, de novo donor-specific antibody formation, graft loss, and patient mortality [8,9,10,11,12,13,14,15,16,17,18,19,20,21]. It has been suggested that IPV of tacrolimus concentrations during the stable phase after transplantation (i.e., beyond six months after KT) could be considered as a tool that may help to identify high-risk patients, and early identification of patients with high IPV may allow for the implementation of appropriate actions to circumvent unwanted clinical outcomes [22].

Among the factors known to influence systemic tacrolimus exposure significantly, the strong impact of a single nucleotide polymorphism involving an A to G transition at position 6986 within intron 3 of the gene-encoding cytochrome P450 (CYP) 3A5 enzyme on between-patient variability of tacrolimus exposure has been consistently reported. CYP3A5 expressers (heterozygous or homozygous carriers of the *CYP3A5*1* allele) display higher tacrolimus clearance and require a higher dose to achieve the same target tacrolimus concentration as CYP3A5 nonexpressers (homozygous carriers of the *CYP3A5*3* allele) [23,24]. Furthermore, drug–drug or drug–food interactions that can interfere with tacrolimus pharmacokinetics, predominantly by inhibiting or inducing CYP3A activity, are well-known as important potential sources of fluctuation of tacrolimus exposure in clinical practice.

The allelic frequency of *CYP3A5*1* and *CYP3A5*3* varies depending on ethnicity. Whereas, only around 15% of Caucasians are CYP3A5 expressers, 45 to 70% of African Americans and approximately 50% of Asians are CYP3A5 expressers [22,23,24]. Limited information regarding the association between the *CYP3A5* genetic polymorphisms and IPV of tacrolimus pharmacokinetics is available [10,25,26,27,28]. The extent to which the *CYP3A5* polymorphisms affect tacrolimus IPV of dose-adjusted tacrolimus trough concentrations (C0/D) has not been explored in a population in which a prevalence of CYP3A5 expressers is high.

To assess whether the *CYP3A5* genetic polymorphisms influence tacrolimus IPV between 6 and 12 months after KT, the primary objective of this study was to compare the IPV of tacrolimus C0/D measured at months 6, 9, and 12 between CYP3A5 expressers and nonexpressers. The secondary objectives were to determine whether tacrolimus C0/D amongst CYP3A5 expressers and nonexpressers change over the study period and whether there is an interaction between the polymorphisms and time after KT on C0/D.

## 2. Materials and Methods

### 2.1. Study Design and Participants

This retrospective cohort study included KTRs (aged >18 years) who underwent KT at the King Chulalongkorn Memorial Hospital (KCMH), a 1500-bed university hospital in Thailand, from January 2011 to June 2017. Only KTRs with a functioning graft at six months post-KT who were on an immediate-release tacrolimus-based immunosuppressive regimen during the 6 to 12 months after transplantation were enrolled.

Patients with the following conditions were excluded: multiple organ transplantation, cirrhosis, hepatitis C viral infection, hepatocellular carcinoma, having biopsy-proven rejection with ongoing treatment that could interfere tacrolimus exposure, having received any investigational drugs or medications (except for prednisolone and proton pump inhibitors) that could significantly interfere tacrolimus pharmacokinetics, and having a record of medication noncompliance. KTRs who passed away or lost kidney grafts before the beginning of this study were also excluded.

This study was conducted in compliance with the Declaration of Helsinki, the Declaration of Istanbul, and the International Conference on Harmonization in Good Clinical Practice Guidelines. It was approved by the Institutional Review Board of the Faculty of Medicine, Chulalongkorn University (IRB number 698/60) and registered in the Thai Clinical Trials Registry (TCTR20180619003). All the patients provided written informed consent before enrolment.

### 2.2. Outcome Measurements

The primary outcome of the analysis was tacrolimus IPV of C0/D estimated by the coefficient of variation (IPVcv). The secondary outcomes were the proportions of KTRs whose IPVcv was 30% or greater based on the US Food and Drug Administration (FDA) criteria for highly within-subject variability of a drug in bioequivalence measures [29], tacrolimus IPV of C0/D estimated by mean absolute deviation method (IPVmad) [8], and C0/D measured at months 6, 9, and 12 after KT. IPV of tacrolimus C0 estimated by percent coefficient of variation (IPVcv C0) and those estimated by mean absolute deviation method (IPVmad C0) were also evaluated between CYP3A5 expressers and nonexpressers.

Because not all KTRs were treated with a stable tacrolimus dose during months 6 to 12 post-KT, C0/D was used for IPV calculation to minimize the effect of dose adjustment on IPV. C0/D was calculated by dividing the measured morning trough concentration of tacrolimus (in ng/mL) by the corresponding daily dose of the drug (in mg/day). To assess the degree of variation, the percentage of coefficient of variation of C0/D, a ratio of standard deviation to the mean of C0/D, was used to quantify the IPVcv [15]: $\left(\sqrt{\sum (\left(\mathrm{Xi}-\mathrm{Xa}{)}^{2}\right)/\left(n-1\right)}\right)\times 100/\;\mathrm{Xa}$, where Xi is the C0/D of tacrolimus (ng/mL per mg/day) measured at 6, 9, and 12 months post-KT, Xa is the average of the three tacrolimus C0/D, and *n* is 3. The percentage of mean absolute deviation, another measure which is less susceptible to outliers, was used for the estimation of IPVmad [8]: $\sum \left(\left|\mathrm{Xi}-\mathrm{Xa}\right|/\;n\right)\times 100/\;\mathrm{Xa}$, where Xi is the C0/D of tacrolimus (ng/mL per mg/day) measured at 6, 9, and 12 months post-KT, Xa is the average of the three tacrolimus C0/D, and *n* is 3.

### 2.3. Data Collection

Demographic and relevant clinical data, including medication lists and tacrolimus levels, were obtained from electronic KCMH medical records and outpatient cards. All participants received immediate-release oral tacrolimus (Prograf^®^, Astellas, Tokyo, Japan) twice daily. There was neither generic substitution nor tacrolimus formulation switching in any of the participants during the study period. Only true C0 obtained during clinically stable conditions and after a stable dosage of the drug was given for at least 14 days were included for the analysis. A stable dosage of prednisolone was prescribed for at least 14 days before tacrolimus C0/D measurement. Data from periods of hospitalization were not included. Episodes of biopsy-proven acute rejection (BPAR) were diagnosed based on histologic findings in indication kidney graft biopsy according to Banff 2015 criteria. Plasma BK virus DNA was determined by quantitative polymerase chain reaction (PCR) assays. Any positive plasma BK virus PCR was reported. Biopsies were evaluated by a transplant pathologist for BK virus-associated nephropathy.

### 2.4. Tacrolimus Concentration Analysis

Whole blood concentrations of tacrolimus were measured at the KCMH laboratory by chemiluminescent microparticle immunoassay (CMIA) method with ARCHITECT system^®^ i1000SR analyzer (Abbott Diagnostic, North Chicago, IL, USA) according to the manufacturer’s protocol [30]. The method was linear in the concentration range of up to 30 ng/mL. The limit of quantification was 0.8 ng/mL. The intra- and inter-assay coefficients of variation for tacrolimus concentrations ranged from 2.9 to 22.4 ng/mL were <10%. According to the manufacturer’s information, the mean recovery of tacrolimus concentrations at 6.9, 9.3, 15.2, and 18.8 ng/mL was 102% (98–107%).

### 2.5. DNA Extraction and Genotyping

Venous blood was drawn from patients when following up at the out-patient visits after recruitment. Total DNA was purified from whole blood using a QIAamp^®^ DNA Blood Mini Kit (QIAGEN, Hilden, Germany) and then stored at −80 °C. A Thermo Scientific NanoDrop 2000c spectrophotometer was used to measure the concentration and purity of DNA samples. The polymorphism of *CYP3A5* was genotyped using TaqMan SNP genotyping assay SNP ID rs776746 (Applied Biosystems^®^, Foster City, CA, USA) and StepOnePlus real-time PCR system (Applied Biosystems^®^, Foster City, CA, USA) for amplifying and detecting specific SNP alleles in genomic DNA samples. The physicians were not aware of the genotype of the patients at the time of any drug dosage modifications.

### 2.6. Statistical Analysis

The distribution of continuous data was evaluated by the Kolmogorov–Smirnov test, and subsequently, parametric or nonparametric tests were applied as appropriate. Normally distributed continuous data are presented as mean ± standard deviation (SD), and non-normally distributed continuous data as medians with interquartile ranges (IQR), unless stated otherwise. Counts and percentages are expressed for categorical data.

Genotype frequencies of the polymorphisms were tested for deviations from Hardy–Weinberg equilibrium using appropriate chi-square testing. The Student’s *t*-test, Mann–Whitney U test, paired *t*-test, or sign test was used to compare continuous data. The chi-square test or Fisher’s exact test was used to compare proportions between groups. Bivariate correlations between continuous variables were assessed by either Pearson’s correlation test or Spearman’s rank correlation test.

To determine the influence of the *CYP3A5* genetic polymorphisms on C0/D over the study period, the impact of the main effects of the polymorphisms (CYP3A5 expressers and nonexpressers) and time (months 6, 9, and 12 post-KT) and the interaction between these on tacrolimus C0/D during the study period were assessed by two-way mixed ANOVA with Bonferroni-corrected post hoc analysis. Assumptions that are required for a test were checked. Where appropriate, data were logarithmically transformed. The homogeneity of variances was evaluated by the Levene test or Mauchly’s test of sphericity. Greenhouse Geisser-corrected values were shown if the assumption of sphericity was violated. Simple main effects for genetic polymorphisms and time after KT were analyzed in the case of a significant interaction, otherwise main effects for genetic polymorphisms and time after KT were interpreted. Only if a significant difference in age, hemoglobin, or serum albumin was found between the genotype groups, were these factors included as covariates in the two-way mixed ANOVA.

Power calculation was performed by G*Power 3.1.9.2 software, Heinrich-Heine-Universität, Düsseldorf, Germany. Based on an assumed effect size of 0.5, a power of 0.8, an estimated dropout rate of 20%, and a significance level of 0.05, a required total sample size of at least 168 KTRs was estimated [31]. All analyses were performed using IBM SPSS statistics 28 (IBM, Bangkok, Thailand). A 2-sided *p* value of less than 0.05 was considered statistically significant.

## 3. Results

A total of 188 adult Thai KTRs participated in this study (Figure 1). Most participants (174 KTRs) received tacrolimus in combination with mycophenolate and prednisolone, while 11 KTRs obtained tacrolimus plus mycophenolate and three received tacrolimus plus prednisolone as their maintenance immunosuppressive regimens.

The median (IQR) of IPVcv was 15.4% (10.2%, 23.9%) and IPVcv of ≥30% was determined in 19 of 188 (10.1%) KTRs, while the median (IQR) of IPVmad was 11.5% (7.6%, 17.9%) and IPVmad of ≥30% was identified in 10 of 188 (5.3%) KTRs. A sign test indicated that IPVcv was significantly higher than IPVmad (*Z* = 13.6, *p* < 0.001, *r* = 0.995) although a very strong positive correlation between IPVcv and IPVmad was observed (Spearman’s correlation coefficient, *r_s_* = 0.997; 95% CI: 0.997 to 0.998; *p* < 0.001), reflecting a perfect monotonic relationship (Figure 2). The overall medians (IQR) of tacrolimus C0/D were 1.38 (0.95, 2.07), 1.37 (1.02, 2.06), and 1.40 (1.00, 2.32) ng/mL per mg/day at months 6, 9, and 12, respectively.

The allele frequency of *CYP3A5*3* was 66.2%. The genotype distribution obeyed the Hardy–Weinberg equilibrium (*X*^2^ = 2.10, *p* = 0.349). The *CYP3A5*1/*1*, **1/*3*, and **3/*3* genotypes were identified in 17 (9.0%), 93 (49.5%), and 78 (41.5%) patients, respectively. Therefore, 110 (58.5%) and 78 (41.5%) patients were classified as CYP3A5 expressers and nonexpressers, respectively.

Demographic and clinical characteristics, including age, hemoglobin, and serum albumin, were comparable between CYP3A5 expressers and nonexpressers (Table 1). Tacrolimus C0/D at months 6, 9, and 12 after KT of CYP3A5 expresser and nonexpresser groups are shown in Figure 3. The distributions of IPVcv of the two groups are depicted in Figure 4. The median (IQR) of IPVcv in 110 CYP3A5 expressers and 78 nonexpressers were 15.8% (10.8%, 23.6%) and 14.5% (10.0%, 23.9%), respectively. No statistically significant difference in IPVcv was found between CYP3A5 expressers and nonexpressers (Mann–Whitney test, *U* = 4104, *p* = 0.613, *r* = 0.037). Proportions of KTRs with their IPVcv of ≥30% were also comparable between CYP3A5 expressers and nonexpressers (10.0% and 10.3%, respectively; Chi-square test, *p* = 0.954, Phi coefficient = 0.004) (Table 2).

A mixed-design ANOVA with a between-patient factor of the *CYP3A5* genetic polymorphisms (CYP3A5 expressers and nonexpressers) and a within-patient factor of time (three time points of tacrolimus concentration monitoring: months 6, 9, and 12 after KT) was performed. Data screening led to the logarithmic transformation of C0/D (logC0/D) to conform data to the normal distribution (Kolmogorov–Smirnov test, all *p* > 0.05). Levene’s test showed an equality of variances for all three time points of observations (all *p* > 0.05). Mauchly’s test indicated that the assumption of sphericity had been violated (*X*^2^(2) = 15.15, *p* < 0.001); therefore, degrees of freedom were corrected using Greenhouse–Geisser estimates of sphericity (ε = 0.927).

The ANOVA revealed that there was no statistically significant interaction between the *CYP3A5* polymorphisms and time for tacrolimus C0/D, *F* (1.85, 345) = 0.279, *p* = 0.740, partial eta-squared = 0.001, suggesting that neither the significant effect of time on the relationship between the *CYP3A5* polymorphisms and logC0/D of tacrolimus nor significant effect of the *CYP3A5* polymorphisms on the relationship between time and logC0/D was detected. However, there was a statistically significant main effect of time, *F* (1.85, 345) = 4.02, *p* = 0.021, with a small effect size (partial eta-squared = 0.021), implying that logC0/D differed minimally but significantly across the three time points. A post hoc pairwise comparison using the Bonferroni correction showed an increased logC0/D from month 6 to month 9 and from month 9 to month 12, but these were not statistically significant (*p* = 0.280 and *p* = 0.616, respectively); however, the increase in logC0/D from month 6 to month 12 was significant (*p* = 0.035). In addition, the main effect of the *CYP3A5* genetic polymorphisms on C0/D was statistically significant, *F* (1, 186) = 158, *p* < 0.001, with a large effect size (partial eta-squared = 0.459), indicating that logC0/D was significantly lower in CYP3A5 expressers than nonexpressers, overall (Table 3).

During months 12 to 24 post-KT, no differences were found in BPAR occurrences, BK viremia detection, and BKVAN development between CYP3A5 expressers and nonexpressers (all *p* > 0.05). Serum creatinine concentrations were also comparable between the two groups at 24 months after KT (*p* = 0.525) (Table 4). The median (IQR) of serum creatinine were comparable between KTRs with IPVcv ≥ 30% and those with IPVcv < 30% (1.4 (1.0, 1.5) vs. 1.3 (1.0, 1.6), 1.2 (1.1, 1.5) vs. 1.2 (1.0, 1.5), and 1.3 (1.0, 1.5) vs. 1.2 (1.0, 1.5) mg/dL) at months 6, 12, and 24, respectively (all *p* > 0.05).

## 4. Discussion

In this cohort study, the impacts of the *CYP3A5* genetic polymorphisms on the IPV of tacrolimus exposure during the 6 to 12 months after KT were assessed in clinically stable adult KTRs who received a stable immediate-release tacrolimus-based immunosuppressive regimen. Of the 188 participants, 110 were CYP3A5 expressers and 78 were nonexpressers, representing 58.5% and 41.5% of the study population, respectively. There was no evidence that the *CYP3A5* genetic polymorphisms significantly influence IPV of tacrolimus exposure during the 6 to 12 months after KT. The medians of IPVcv were comparable between CYP3A5 expressers and nonexpressers (15.8% vs. 14.5%, respectively). Moreover, when considering a ≥30% within-subject variability cut point, which is in accordance with the US FDA definition of highly variable drugs, similar proportions of KTRs with IPVcv of at least 30% were determined between CYP3A5 expressers and nonexpressers (10.0% vs. 10.3%, respectively). IPVmad, IPVcv C0, and IPVmad C0 were also comparable.

Given the fact that CYP3A5 expressers require a dose of tacrolimus approximately twice as high as CYP3A5 nonexpressers to achieve the same tacrolimus levels, it has been hypothesized that CYP3A5 expressers could be more susceptible to higher IPV [32]. On the other hand, it has also been postulated that since tacrolimus metabolism in CYP3A5 nonexpressers relies predominantly on the activity of CYP3A4 which is much more prone to induction and inhibition [33], CYP3A5 nonexpressers might be more susceptible to higher IPV [28]. The results of the present study do not support these hypotheses. With different parameters and methods used to quantify IPV, all findings of this study consistently show that there is no direct evidence of the significant influence of the *CYP3A5* genetic polymorphisms on IPV of tacrolimus exposure in a stable clinical period.

In agreement with our findings, it has been observed that CYP3A5 expression is not associated with IPV of clearance, absolute concentration, or predicted area under the concentration-time curve (AUC) of tacrolimus among stable adult KTRs in previous studies [10,25,26,27,28]. In two studies by Pashaee et al. [25] and Spierings et al. [26], the proportions of KTRs with IPV of relative clearances of tacrolimus lower than the median values observed were comparable between CYP3A5 expressers and nonexpressers. The IPV of absolute concentrations of tacrolimus did not differ significantly between CYP3A5 expressers and nonexpressers in two other studies by Ro et al. [10] and Muller et al. [28]. In addition, the IPV of tacrolimus AUC estimated by a limited sampling strategy did not differ significantly between CYP3A5 expressers and nonexpressers in a previous study by Cheung et al. [27].

Despite high tacrolimus IPV being recognized as a risk factor for poor kidney transplant outcomes, a standardized method for the determination of tacrolimus IPV in kidney transplantation has not been established. Different methods have been used to quantify IPV of tacrolimus among studies that reported the associations between high tacrolimus IPV and poor long-term outcomes. The coefficient of variation and mean absolute deviation are commonly utilized to calculate tacrolimus IPV [8,9,10,11,12,13,14,15,16,18,19,20,21]. A minimum of three C0 concentrations have been included in the calculation of tacrolimus IPV for each patient among studies [8,12,14,15]. Considering that C0/D should be determined in a stable clinical situation with a sufficient number of C0 available in a period in which early interventions can be provided, the IPV of tacrolimus C0/D calculated from at least three C0 values measured during 6–12 months after transplantation has been suggested to be utilized as a marker for clinical outcomes [32].

Alongside IPVcv, IPVmad was also determined in the present study. The medians of IPVcv and those of the IPVmad were comparable between CYP3A5 expressers and nonexpressers. The proportions of KTRs with ≥30% IPVcv and those with ≥30% IPVmad were similar between the two groups. Even though a very strong positive correlation between IPVcv and IPVmad was determined (*r_s_* = 0.997), the medians of IPVcv and IPVmad were significantly different. Twice KTRs with IPV ≥30% were identified by IPVcv as by IPVmad in the present study, implying that different cutoffs of these parameters may be needed to identify a patient at risk for poor clinical outcomes.

Although a significant influence of the *CYP3A5* genetic polymorphisms on IPV of tacrolimus was not detected in the present study, the significant impact of the polymorphisms on C0/D during months 6 to 12 was confirmed. Tacrolimus is extensively metabolized by the CYP3A5 enzyme, which is polymorphically expressed. The *CYP3A5* genetic polymorphism is a well-known factor accounting for a substantial portion of the variable pharmacokinetics of tacrolimus [34,35,36]. The single nucleotide polymorphism involves an A to G transition at position 6986 within intron 3 of the gene-encoding CYP3A5 resulting in an alternative splice variant with an early stop codon that generates a non-functional protein [24,34]. Individuals who are heterozygous or homozygous carriers of the *CYP3A5*1* allele (CYP3A5 expressers) produce high levels of full-length *CYP3A5* mRNA and express high levels of functional CYP3A5 protein while homozygous carriers of the *CYP3A5*3* allele (CYP3A5 nonexpressers) have very low or undetectable levels of functional CYP3A5 proteins.

In the same direction as our findings, an approximately 1.7 times lower tacrolimus C0/D was observed in CYP3A5 expressers compared with nonexpressers during the early period post-KT in a previous study by Phupradit et al. [36]. In addition, it has been reported that tacrolimus C0/D was significantly lower in CYP3A5 expressers compared with nonexpressers in the overall analysis and when stratifying for ethnicity or time after transplantation (≤1, 3–6, 12–24 months), despite the presence of significant heterogeneity in all comparisons in a meta-analysis by Terrazzino et al. [37].

Not only do *CYP3A5* genetic polymorphisms impact tacrolimus concentrations in the systemic circulation, but they also may influence a patient’s susceptibility to tacrolimus drug interactions. The present study was performed in stable KTRs, not receiving any potent interacting medication; whether drug interactions would differently affect tacrolimus IPV amongst CYP3A5 expressers and nonexpressers remains to be further investigated.

A small but significant increase in the overall C0/D from month 6 to month 12, observed in the present study, is supported by a previous population pharmacokinetic study which demonstrated that apparent clearance of tacrolimus decreases with increasing duration of tacrolimus therapy [38]. Notably, the slope of apparent clearance reduction was substantially different between the early and late phases of tacrolimus therapy. While tacrolimus apparent clearance decreases precipitously during the early months of treatment, it declines gradually after approximately six months of treatment [38]. A reduction in steroid dosage, decreased CYP3A4 activity, and increased hematocrit and albumin with time are plausible explanations for the decline of tacrolimus apparent clearance after KT [5,39]. Timing after KT during which tacrolimus exposure is used for IPV calculation is, therefore, one possible explanation for the differences in IPV values reported among previous studies [8,9,10,11,12,13,14,15,16,17,18,19,20,21].

The influence of other SNPs, including *CYP3A4*22* (rs35599367), *CYP3A5*6* (rs10264272), and *CYP3A5*7* (rs41303343), on tacrolimus exposure, has also been reported in previous studies [40,41]. Given that less than 1% of the frequency of these SNPs was found among Asian populations [22], the effect of these SNPs was not evaluated in the study patients. Whether these SNPs would impact tacrolimus IPV is questionable.

The present study has several strengths. In this study, IPV was determined only in a stable clinical period. The medication records were checked to ensure that the patients had not received any medications that could significantly interfere with tacrolimus pharmacokinetics, with the exceptions of proton pump inhibitors and prednisolone. In addition, since the number of C0 values used for the calculation can affect IPV, tacrolimus IPV calculated for each individual patient in this study was based on three measurements at months 6, 9, and 12. Moreover, the IPV determined by different methods commonly reported in clinical studies were provided.

The present study has limitations that should be considered. First, this was a single-center, retrospective study; however, the study patients were uniformly treated with the same strategies for caring for the patients post-KT. Second, tacrolimus concentrations were measured by CMIA method. Although liquid chromatography-tandem mass spectrometry is available in large transplant institutions, immunoassays are still used in many centers for the therapeutic monitoring of tacrolimus. Additionally, this study focused on the variability of tacrolimus exposure during 6 to 12 months post-KT; the study did not have sufficient power to detect any difference (if one existed) of clinical outcomes between the patients with the IPV of ≥30% and <30%.

## 5. Conclusions

No evidence of a difference in IPV of tacrolimus C0/D between CYP3A5 expressers and nonexpressers was found in clinically stable conditions. The two genotype groups determined comparable medians of tacrolimus IPVcv, IPVmad, and proportions of KTRs with the IPVcv ≥ 30% in the present study. Tacrolimus C0/D in CYP3A5 expressers was approximately half as high as those in nonexpressers at all three study time points during 6 to 12 months post-KT. Small but significant increases in tacrolimus C0/D from month 6 to month 12 were observed in both groups of CYP3A5 expressers and nonexpressers.

## Figures and Tables

**Figure 1 jcm-11-06320-f001:**
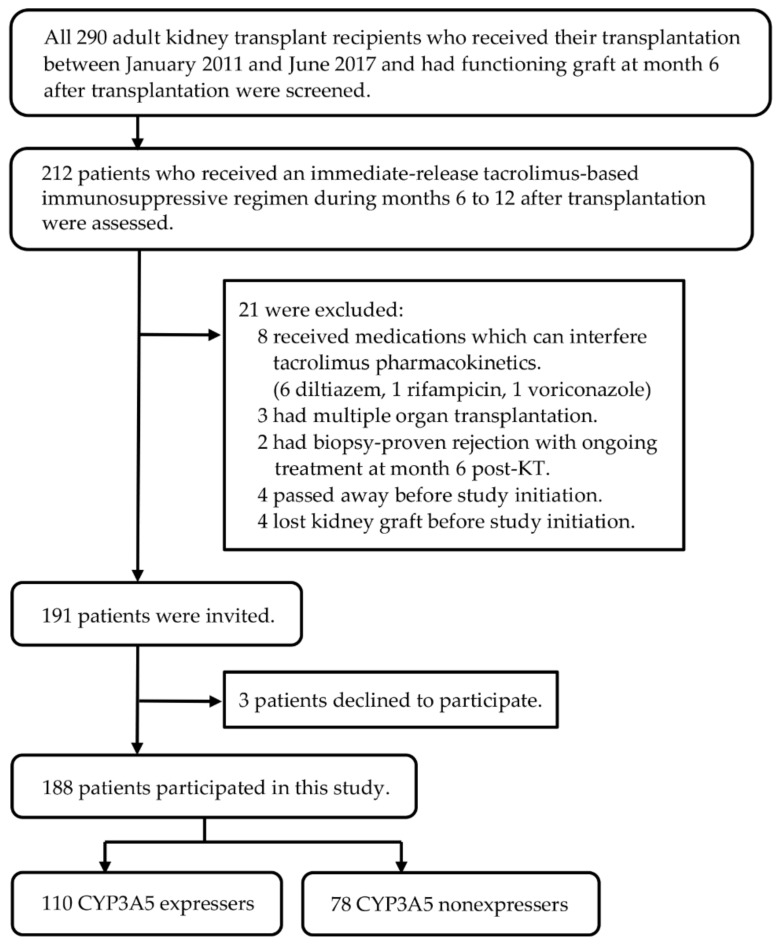
Flow diagram of the inclusion and exclusion process.

**Figure 2 jcm-11-06320-f002:**
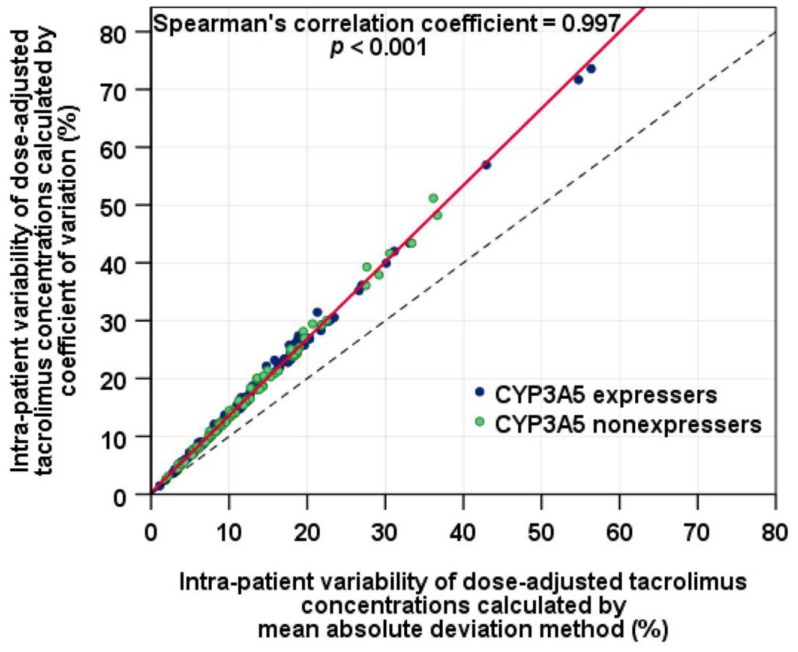
Correlation of intra-patient variability of tacrolimus dose-adjusted trough concentrations estimated by coefficient of variation and intra-patient variability of tacrolimus dose-adjusted trough concentrations estimated by mean absolute deviation method (the solid line is the regression line; the dashed line represents the line of identity).

**Figure 3 jcm-11-06320-f003:**
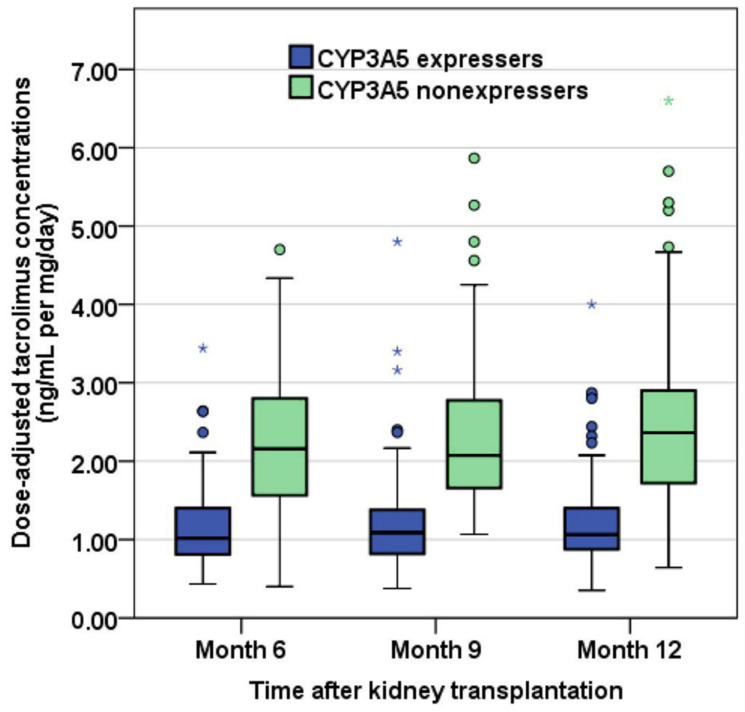
Box plots of dose-adjusted tacrolimus trough concentrations in CYP3A5 expressers and nonexpressers at months 6, 9, and 12 post-kidney transplantation (the circle is the value that exceeds the first quartile minus 1.5 times the interquartile range or the third quartile plus 1.5 times the interquartile range; the asterisk represents the value that exceeds the first quartile minus 3.0 times the interquartile range or the third quartile plus 3.0 times the interquartile range).

**Figure 4 jcm-11-06320-f004:**
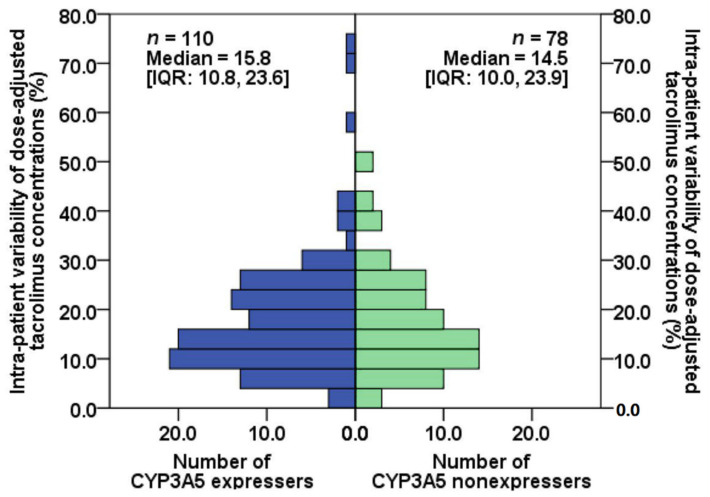
Frequency distributions of intra-patient variability of tacrolimus dose-adjusted trough concentrations estimated by coefficient of variation in CYP3A5 expressers and nonexpressers.

**Table 1 jcm-11-06320-t001:** Patient characteristics.

Characteristics	CYP3A5 Expressers (*n* = 110)	CYP3A5 Nonexpressers (*n* = 78)	*p*-Value
**On the day of transplantation**			
Recipient age, years	45.8 ± 11.7	44.7 ± 11.6	0.538 ^b^
Body weight, kg	58.2 ± 11.2	58.0 ± 11.7	0.889 ^b^
Female, *n* (%)	49 (44.5)	35 (44.9)	0.965 ^a^
Previous KT, *n* (%)	7 (6.4)	3 (3.8)	0.527 ^c^
Panel reactive antibody > 20%, *n* (%)	19 (17.3)	12 (15.4)	0.731 ^a^
Human leukocyte antigen mismatches, no.	3.0 [2.0, 4.0]	3.0 [2.0, 4.0]	0.973 ^d^
Deceased donor, *n* (%)	56 (50.9)	36 (46.2)	0.520 ^a^
Donor age, years	36.5 [28.8, 46.2]	38.0 [28.2, 46.2]	0.723 ^d^
Cold ischemic time, minutes	253.5 [20.8, 1078.5]	68.0 [21.5, 1047.0]	0.907 ^d^
Renal replacement therapy before KT, *n* (%)	-	-	0.234 ^c^
Preemptive transplantation	2 (1.8)	4 (5.1)	-
Hemodialysis	97 (88.2)	70 (89.7)	-
Peritoneal dialysis	11 (10.0)	4 (5.1)	-
**At month 6 post-kidney transplantation**			
Serum creatinine, mg/dL	1.3 [1.0, 1.6]	1.2 [1.0, 1.5]	0.377 ^d^
Hemoglobin, g/dL	12.9 ± 1.8	13.2 ± 1.8	0.353 ^b^
Serum albumin, g/dL ^e^	4.3 [4.1, 4.6]	4.3 [4.2, 4.6]	0.495 ^d^
**At month 9 post- kidney transplantation**			
Serum creatinine, mg/dL	1.3 [1.0, 1.5]	1.2 [1.0, 1.5]	0.475 ^d^
Hemoglobin, g/dL	13.2 ± 1.8	13.5 ± 1.8	0.272 ^b^
Serum albumin, g/dL ^f^	4.4 [4.2, 4.6]	4.4 [4.2, 4.6]	0.742 ^d^
**At month 12 post- kidney transplantation**			
Serum creatinine, mg/dL	1.3 [1.0, 1.5]	1.2 [1.0, 1.5]	0.327 ^d^
Hemoglobin, g/dL	13.4 ± 1.7	13. 6 ± 1. 7	0.421 ^b^
Serum albumin, g/dL ^g^	4.4 [4.2, 4.6]	4.4 [4.3, 4.7]	0.527 ^d^

Continuous values are expressed as mean ± SD or median [interquartile range] for comparison reasons. ^a^ Chi-square test. ^b^ Student’s *t*-test. ^c^ Fisher’s exact test. ^d^ Mann–Whitney *U* test. ^e^ Serum albumin at month 6 was that of 137 patients only. ^f^ Serum albumin at month 9 was that of 128 patients only. ^g^ Serum albumin at month 12 was that of 125 patients only.

**Table 2 jcm-11-06320-t002:** Tacrolimus exposure during months 6 to 12 post-kidney transplantation.

Tacrolimus Exposure ^a^	CYP3A5 Expressers (*n* = 110)	CYP3A5 Nonexpressers (*n* = 78)	*p*-Value
**Month 6 post-kidney transplantation**			
C0, ng/mL	6.6 [5.8, 7.8]	6.7 [5.3, 8.2]	0.747 ^b^
Dose, mg/day	6.0 [5.0, 8.1]	3.0 [2.5, 4.5]	<0.001 ^b^
C0/D, ng/mL per mg/day	1.02 [0.80, 1.40]	2.16 [1.55, 2.82]	<0.001 ^b^
**Month 9 post-kidney transplantation**			
C0, ng/mL	6.2 [5.5, 7.6]	6.4 [5.4, 7.8]	0.708 ^b^
Dose, mg/day	6.0 [4.5, 8.0]	3.0 [2.0, 4.0]	<0.001 ^b^
C0/D, ng/mL per mg/day	1.09 [0.82, 1.38]	2.07 [1.65, 2.80]	<0.001 ^b^
**Month 12 post-kidney transplantation**			
C0, ng/mL	6.2 [5.3, 7.1]	6.6 [5.6, 7.8]	0.161 ^b^
Dose, mg/day	5.5 [4.5, 7.0]	3.0 [2.0, 4.0]	<0.001 ^b^
C0/D, ng/mL per mg/day	1.06 [0.88, 1.40]	2.36 [1.72, 2.90]	<0.001 ^b^
**Intra-patient variability**			
IPVmad C0, %	12.6 [7.6, 19.2]	14.4 [9.6, 20.0]	0.193 ^b^
IPVcv C0, %	17.7 [10.0, 25.9]	19.7 [12.8, 26.5]	0.176 ^b^
IPVmad, %	11.6 [7.9, 17.8]	10.8 [7.5, 17.9]	0.686 ^b^
Number of patients with IPVmad ≥ 30%, *n* (%)	6 (5.5)	4 (5.1)	1.000 ^c^
IPVcv, %	15.8 [10.8, 23.6]	14.5 [10.0, 23.9]	0.613 ^b^
Number of patients with IPVcv ≥ 30%, *n* (%)	11 (10.0)	8 (10.3)	0.954 ^d^

C0, tacrolimus trough concentration; C0/D, dose-adjusted tacrolimus trough concentration; IPVcv, intra-patient variability of tacrolimus exposure estimated by coefficient of variation of dose-adjusted tacrolimus trough concentration measured at months 6, 9, and 12 post-kidney transplantation; IPVcv C0, intra-patient variability of tacrolimus exposure estimated by coefficient of variation of tacrolimus trough concentration measured at months 6, 9, and 12 post-kidney transplantation; IPVmad, intra-patient variability of tacrolimus exposure estimated by mean absolute deviation of dose-adjusted tacrolimus trough concentration measured at months 6, 9, and 12 post-kidney transplantation; IPVmad C0, intra-patient variability of tacrolimus exposure estimated by mean absolute deviation of tacrolimus trough concentration measured at months 6, 9, and 12 post-kidney transplantation. ^a^ Continuous values are expressed as median [interquartile range]. ^b^ Mann–Whitney test. ^c^ Fisher’s exact test. ^d^ Chi-square test.

**Table 3 jcm-11-06320-t003:** Two-way mixed analysis of variance results for the logarithm of dose-adjusted tacrolimus trough concentrations.

LogC0/D	*n*	Month 6	Month 9	Month 12	Total Means	95% CI ^a^
CYP3A5 expressers	110	0.03 ± 0.18	0.04 ± 0.17	0.05 ± 0.17	0.04 ^b^	0.01, 0.07
CYP3A5 nonexpressers	78	0.32 ± 0.18	0.34 ± 0.17	0.36 ± 0.19	0.34 ^b^	0.30, 0.38
**Total means**	-	0.18 ^c^	0.19	0.20 ^c^	-	-
**95% CI ^a^**	-	0.15, 0.20	0.17, 0.22	0.18, 0.23	-	-
**Source**	**Sum of Squares**	**Mean square**	** *df* **	** *F* **	***p*-Value**	**Partial eta-squared**
*CYP3A5*3* polymorphisms	12.2	12.2	1, 186	158	<0.001	0.459
Time	0.072	0.039	1.85, 345	4.02	0.021	0.021
Polymorphisms × Time	0.005	0.003	1.85, 345	0.279	0.740	0.001

LogC0/D, logarithms of dose-adjusted trough concentrations of tacrolimus in ng/mL per mg/day. ^a^ 95% confidence interval of total means. ^b^ *p* < 0.001 for comparison between CYP3A5 expressers and nonexpressers. ^c^ *p* < 0.05 for comparison between month 6 and month 12 (Bonferroni correction).

**Table 4 jcm-11-06320-t004:** Biopsy-proven acute rejection, BK viral infection, and serum creatinine at 24 months post-kidney transplantation.

During Months 12–24 Post-Kidney Transplantation	CYP3A5 Expressers (*n* = 110)	CYP3A5 Nonexpressers (*n* = 78)	*p*-Value
**Overall BPAR occurrence, *n* (%)**	5 (4.5)	2 (2.6)	0.701 ^a^
Acute cellular rejection, *n* (%)	2 (1.8)	0 (0.0)	0.512 ^a^
Antibody-mediated rejection, *n* (%)	3 (2.7)	2 (2.6)	1.000 ^a^
**BK viremia detection, *n* (%)**	6 (5.5)	3 (3.8)	0.738 ^a^
Plasma BK viral load of ≥10,000 copies/mL, *n* (%)	2 (1.8)	0 (0.0)	0.512 ^a^
**BKVAN development, *n* (%)**	2 (1.8)	0 (0.0)	0.512 ^a^
**Serum creatinine at month 24, mg/dL** ^c^	1.3 [1.0, 1.5]	1.2 [1.0, 1.5]	0.525 ^b^

BPAR, biopsy-proven acute rejection; BKVAN, BK virus-associated nephropathy. ^a^ Fisher’s exact test. ^b^ Mann–Whitney test; ^c^ Serum creatinine are presented as median [interquartile range].

## Data Availability

The data are available from the corresponding author upon reasonable request.

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
