# Peer review of "Comparison of Tacrolimus Intra-Patient Variability during 6–12 Months after Kidney Transplantation between CYP3A5 Expressers and Nonexpressers"

_jcm, 2022, doi:10.3390/jcm11216320_

Round 1
Reviewer 1 Report
Dear Dr. Nuchjumroon,
your paper addresses an important aspect in transplantation medicine, since intra-patient tacrolimus variability (IPV) is more and more characterized as an important impact factor for resulting allograft function. However, several aspects should be addressed to enhance significance and novelty of the paper, which is currently somehow limited:
- - as different publications found sometimes contradictory statements a discussion/statement on the possible impact of ethnicity regarding CYP3A5 expression and impact regarding tacrolimus IPV might to interesting;
- - as steroid-dosage may influence tacrolimus C/D ratio: have the patients been on stable steroid dosage at least 7 days before C/D measurement;
- - did the authors check for different anti-HLA antibody development between expressers and non-expressers? Or allograft rejection? expression of opportunistic viral infections, e.g. BKV
- - have other CYP3A5 alleles been checked e.g. CYP3A5*6.
Reviewer 2 Report
This well written article describes the pharmacokinetics of tacrolimus in kidney transplant recipients. The authors describe the characteristics of tacrolimus use in patients who are CYP3A5 expressers and non expressers. They found that IPV does not differ based on CYP3A5 status.
Overall this is an interesting article with sound methodology. I have minor concerns including the following:
1. Can the authors explain why patients who developed BPAR within the first 6 months of transplant were excluded. I would want to especially understand the pharmacokinetics of tacrolimus in these patients given this important clinical outcome and its association with IPV.
2. Furthermore, what were the incidences of BPAR or BK viremia between expressers and non expressers. These are very important clinical outcomes.
3. When was the CYP3A5 genotype performed?
Round 2
Reviewer 1 Report
Dear Dr. Nuchjumroon,
you addressed the requested remarks in an appropriate manner.